# Natural mismatch repair mutations mediate phenotypic diversity and drug resistance in *Cryptococcus deuterogattii*

R Blake Billmyre[†], Shelly Applen Clancey, Joseph Heitman*

Department of Molecular Genetics and Microbiology, Duke University School of Medicine, Duke University Medical Center, Durham, United States

**Abstract** Pathogenic microbes confront an evolutionary conflict between the pressure to maintain genome stability and the need to adapt to mounting external stresses. Bacteria often respond with elevated mutation rates, but little evidence exists of stable eukaryotic hypermutators in nature. Whole genome resequencing of the human fungal pathogen *Cryptococcus deuterogattii* identified an outbreak lineage characterized by a nonsense mutation in the mismatch repair component *MSH2*. This defect results in a moderate mutation rate increase in typical genes, and a larger increase in genes containing homopolymer runs. This allows facile inactivation of genes with coding homopolymer runs including *FRR1*, which encodes the target of the immunosuppressive antifungal drugs FK506 and rapamycin. Our study identifies a eukaryotic hypermutator lineage spread over two continents and suggests that pathogenic eukaryotic microbes may experience similar selection pressures on mutation rate as bacterial pathogens, particularly during long periods of clonal growth or while expanding into new environments.

DOI: https://doi.org/10.7554/eLife.28802.001

*For correspondence:
heitm001@duke.edu

Present address: [†]Stowers Institute for Medical Research, Kansas City, United States

Competing interests: The authors declare that no competing interests exist.

## Introduction

Mutation is the raw material of evolution. As a result, all organisms must strike a balance between allowing enough random mutations for selection to act upon, and the fact that most of these mutations are likely to be deleterious and must be purged from the population. This is particularly true for pathogenic microbes that take part in Red Queen conflicts with their hosts and require a continuous supply of mutations to remain competitive. In this case, increased pressure may exist to maintain an elevated mutation rate to increase the rate of adaptation. Increases in mutation rate may also serve to accelerate adaptation when microbes are introduced into new environments or encounter novel stresses, such as antimicrobial therapy.

Adaptive variation in mutation rate is common in bacteria. One example is the Long Term Evolution Experiment (LTEE), where defects in DNA repair emerged and the resulting hypermutator phenotype swept the population in six out of twelve *E. coli* lines (*Tenaillon et al., 2016*). Mutator alleles likely emerge frequently but are typically purged from the population because individual mutations are more likely to be deleterious than adaptive; as a result hypermutators generally produce less fit offspring than non-hypermutators. However, occasionally a sufficiently beneficial mutation is potentiated by the presence of the hypermutator and the hypermutator allele is able to hitchhike to higher frequency within the population. This is more likely to occur in populations at local evolutionary minima, where many different large-effect adaptive mutations are possible. For example, hypermutator phenotypes are common in isolates of *Pseudomonas aeruginosa* growing within the lungs of patients with Cystic Fibrosis, where environmental conditions are constantly changing (*Oliver et al., 2000*). This changing environment causes an ongoing need to adapt, and increases the likelihood of hypermutators emerging by increasing the target size for adaptive variation. However, once beneficial

**eLife digest** As humans, we often think of genetic mutations as being bad. Over the past several decades we have seen health warnings issued on a variety of environmental exposures, from cigarettes to tanning beds, and with good reason because they cause mutations. For multicellular organisms like humans, these mutations are strongly associated with cancer. But in bacteria, this is not true. In fact, the rate at which mutations occur sometimes increases to help bacteria cope with stressful environments.

Unlike bacteria, humans are eukaryotes – the name given to organisms whose cells contain different compartments separated by membranes, such as the nucleus of the cell. For years, we have assumed that eukaryotic microbes, like fungi and parasites, act more like humans than like bacteria because work in budding yeast (another eukaryote) has suggested this to be the case. However, recent work in disease-causing fungi has shown that, much like bacteria, elevated mutation rates may help them to respond to stress. This could also enable fungi to become resistant to drugs used to treat fungal infections.

*Cryptococcus deuterogattii* is a fungus that causes human diseases including meningoencephalitis and a lung infection called pulmonary cryptococcosis. An ongoing outbreak of the fungus began in the Pacific Northwest of Canada in the late 1990s and emerged in the United States in 2006/2007. Among isolates closely related to those fungi causing the outbreak, three were found that appear to have a specific mutation in their DNA mismatch repair pathway, meaning that they may also experience a higher mutation rate. These strains are also less able to cause disease than others.

Billmyre et al. now demonstrate experimentally that all three isolates have a specific DNA mismatch repair defect, and show that these fungi experience elevated mutation rates, resulting in what is known as a hypermutator state. Furthermore, whole genome sequencing and phylogenetic analysis showed that these hypermutator strains are derived from the outbreak-causing fungi, and that their reduced ability to cause disease is likely a result of accumulating mutations and the loss of the ability to grow at the higher temperatures found in the human body.

Fungal infections are difficult to treat, in part because there are a limited number of available drugs. Elevated mutation rates will likely increase how often and how rapidly fungi develop resistance to these drugs. Understanding how commonly fungi exhibit a hypermutator state that could impact the development of drug resistance will therefore be important for treating patients with fungal infections, which account for millions of infections and hundreds of thousands of deaths annually worldwide.

DOI: https://doi.org/10.7554/eLife.28802.002

mutations have fixed, and organisms have adapted to their new environment, antimutator suppressor alleles can emerge to reduce the mutation rate or defective mutator alleles can even be replaced by functional alleles through horizontal gene transfer (*Wielgoss et al., 2013*; *Denamur et al., 2000*). This balance is likely critical to evolution in microbes as they face diverse stresses and environmental changes, and as a result over 1% of natural bacterial isolates display hypermutator phenotypes (*LeClerc et al., 1996*). Even more extreme cases also exist, like in *Mycoplasma pneumoniae* and *M. genitalium*, where the mismatch repair machinery has been completely lost (*Himmelreich et al., 1997*).

In fact, many mutator phenotypes are the result of defects in mismatch repair, commonly referred to as MMR. In bacteria, MMR requires the MutHLS system, where MutS binds to mismatches and recruits MutL, which subsequently activates MutH to excise the mispair (*Su and Modrich, 1986*; *Au et al., 1992*; *Grilley et al., 1989*; *Cox et al., 1972*). In eukaryotes, both the MutS and MutL families have expanded, yielding MutS homolog (MSH) (*Reenan and Kolodner, 1992*) and MutL homolog (MLH) families (*Strand et al., 1993*). Heterodimers of various MSH family members participate in mismatch repair, but Msh2 in particular is a core component of the majority of mismatch repair pathways in yeast (*Harfe and Jinks-Robertson, 2000*). As a result, defects in MMR, and *MSH2* mutations in particular, result in elevated rates of simple mismatches, but also dramatically elevated rates of repeat tract instability (*Strand et al., 1993*), which are associated with hereditary colon cancer in humans. MMR also plays a role in rejecting heteroduplexes of divergent sequences during both

mitosis (*Datta et al., 1997*) and meiosis (*Alani et al., 1994*), meaning that loss of the complex can lower species boundaries and allow increased recombination between divergent chromosomes (*Alani et al., 1994*; *Hunter et al., 1996*; *Rayssiguier et al., 1989*).

In contrast to bacteria, substantially less is known about hypermutators in natural populations of eukaryotes. Evolutionary theory predicts that the presence of sex abrogates selection for hypermutator strains by eliminating genetic linkage between the mutator allele and the associated beneficial mutations it hitchhikes upon to high frequency (*Tenaillon et al., 2000*). Studies of allele incompatibility within the model fungus *Saccharomyces cerevisiae* have supported this traditional paradigm. Incompatible alleles of *PMS1* and *MLH1* exist within the population, such that a cross of two strains with wildtype mutation frequency can give rise to progeny with elevated mutation rates (*Heck et al., 2006*; *Bui et al., 2015*). However, the homozygous incompatible arrangement was not found among any of the original 65 strains examined, and was only identified within one clinical isolate of 1010 global diverse yeast isolates (*Bui et al., 2017*) and four clinical strains out of 93 sequenced in a separate study (*Skelly et al., 2017*). This frequency is far below that of the individual alleles in the population, and in each case observed suppressors had arisen to restore a wildtype mutation rate despite the incompatibility. In contrast, eukaryotic mutators can adapt and thrive in clonally expanding populations such as in human tumor environments (*Bielas et al., 2006*). Recent studies have begun to challenge this dogma in eukaryotic microorganisms as well, with over 50% of clinical *C. glabrata* isolates harboring loss of function mutations in *MSH2*, resulting in loss of mismatch repair function (*Healey et al., 2016*). Two groups also recently independently identified hypermutators in the fungal pathogen *Cryptococcus neoformans*. In the first case, long branch lengths were observed in genome sequencing of serial isolates from a recurrent infections that are correlated with nonsense mutations in *MSH2*, *MSH5*, and *RAD5 Rhodes et al., 2017*.In the second case, two of eleven clinical isolates exhibited elevated mutation rate that is linked to either an *MSH2* nonsense mutation or multiple *MSH2* missense mutations (*Boyce et al., 2017*). In previous work we also identified a clade of candidate hypermutators within the Pacific Northwest *Cryptococcus deuterogattii* outbreak that contain a coding single base deletion within the critical mismatch repair component *MSH2* (*Billmyre et al., 2014*).

*Cryptococcus deuterogattii*, is a basidiomycete human fungal pathogen previously known as *Cryptococcus gattii* VGII and recently elevated to species level (*Hagen et al., 2015*), although this decision is disputed by a portion of the community (*Kwon-Chung et al., 2017*; *Hagen et al., 2017*). Unlike the other species of the *Cryptococcus* pathogenic species complex that infect immunocompromised hosts, *C. deuterogattii* is characterized by its ability to cause infection in otherwise healthy hosts (*Springer et al., 2012*) and by loss of the RNAi pathway (*Feretzaki et al., 2016*; *D'Souza et al., 2011*). *C. deuterogattii* is responsible for an ongoing outbreak in the Pacific Northwest region of the United States and Canada, which began in the late 1990's (*Kidd et al., 2004*). This outbreak was originally analyzed using Multi-locus sequence typing (MLST) and shown to be comprised of three clonal expansions, denoted VGIIa, VGIIb, and VGIIc (*Fraser et al., 2005*; *Byrnes et al., 2010*). These three subtypes differ in terms of virulence and total number of cases, with VGIIa responsible for the majority of infections. More recently, whole genome sequencing studies of these subpopulations showed that the clonal subtypes appeared to have different proximal geographic origins, and most interestingly, that VGIIa has three very closely related isolates, here referred to as VGIIa-like, that were considerably less virulent and differed in isolation location or date from the outbreak (*Billmyre et al., 2014*). We identified a candidate potential large effect mutation shared by these three diminished virulence isolates that resulted in a predicted non-functional Msh2. Here we show that these VGIIa-like isolates exhibit a *bona fide* hypermutator phenotype. Furthermore, we show that homopolymer runs are particularly unstable, and that these runs are common within the coding regions of genes in the *Cryptococcus* genome. We predict that this results in dramatic phenotypic diversity from inactivation and possibly also activation or reactivation of genes. Finally, we show that the mutant *msh2* allele is not directly responsible for the decrease in virulence of the VGIIa-like isolates, nor does it appear to have directly played a role in the evolution of virulence in the VGIIa clonal outbreak. Rather, it appears to represent a parallel route of adaptation to a new environment in a successful lineage isolated from two continents over two decades and from both the environment and patient samples. This work suggests that hypermutator phenotypes are not limited to prokaryotes, but may represent a frequent avenue to evolutionary change and phenotypic diversity in eukaryotic microbes as well.

## Results

We previously identified a sublineage of strains highly related to the VGIIa outbreak in the Pacific Northwest denoted VGIIa-like, in part because they differed either in isolation location or time from the typical outbreak isolates (*Billmyre et al., 2014*). The VGIIa-like sublineage is composed of three isolates: NIH444, a clinical isolate from Seattle in 1975, well before the start of the outbreak; ICB107, a clinical isolate from Brazil in 1981; and CBS7750, an environmental isolate from California in 1990. All three isolates are characterized by diminished virulence relative to the VGIIa clonal outbreak strains (*Fraser et al., 2005*; *Byrnes et al., 2010*; *Hagen et al., 2013*). The analysis here includes additional genomes for a South American outgroup to both the VGIIa-like and VGIIa groups (*Engelthaler et al., 2014*)(*Figure 1A*) as well as alignment of these genomes to the improved *C. deuterogattii* genome assembly (*Farrer et al., 2015*). This allowed determination of the ancestral state of the multiple alleles differentiating the VGIIa and VGIIa-like groups. We also generated a second independent genome sequence for our original NIH444 isolate (NIH444(1) and NIH444(2)), as well as an NIH444 isolate acquired from a different lab (NIH444(v)) (*Springer et al., 2010*). We identified a total of 19 mutations that are shared by all of the VGIIa-like isolates sequenced: 13 SNPs and 6 INDELs (*Figure 1—source data 1*). Five SNPs are noncoding while 7 of the 8 remaining SNPs were nonsynonymous mutations. In addition, we confirmed that one of the INDELs in this branch is a single base deletion in a coding exon of *MSH2* (26). This deletion is unique to the VGIIa-like group.

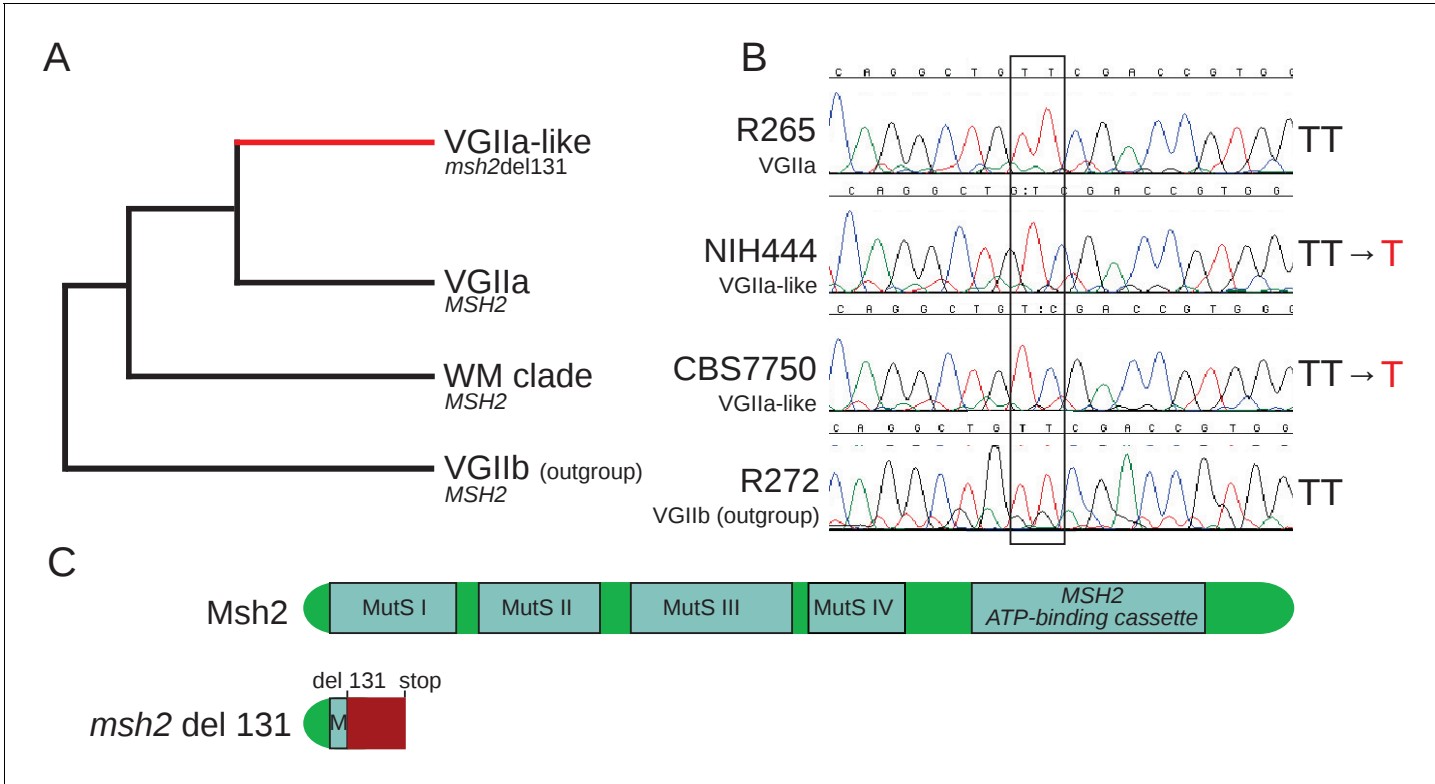

**Figure 1.** VGIIa-like isolates harbor a frameshift resulting in a predicted nonfunctional protein. (**A**) The VGIIa-like clade is part of the clonal radiation that includes the Pacific Northwest Outbreak and shares with the VGIIa clade three ancestral isolates from South America described here as the WM clade. All three isolates are characterized by a frameshift in the *MSH2* gene at position 131. (**B**) Sanger sequencing of the *MSH2* gene confirms that NIH444 and CBS7750 have both undergone deletion of a single T within the coding region of *MSH2*, while the VGIIa R265 strain and the outgroup VGIIb strain have not. (**C**) This deletion results in a frameshift beginning in the first functional domain of *MSH2* and an early premature stop. This truncated protein is predicted to be non-functional.

DOI: https://doi.org/10.7554/eLife.28802.003

The following source data is available for figure 1:

**Source data 1.** Mutations shared by VGIIa-like strains relative to VGIIa.

DOI: https://doi.org/10.7554/eLife.28802.004

We confirmed both the presence of this mutation and showed that it results in a hypermutator phenotype. Sanger sequencing verified that this single base deletion is not an Illumina sequencing artifact (*Figure 1B*). This single base deletion occurs within an early exon of *MSH2* and results in a nonsense mutation, with both an out of frame message and an early stop codon present in the predicted transcript (*Figure 1C*).

## *MSH2* defect results in hypermutator phenotype

Because Msh2 is a critical component of the mismatch repair complex, we next tested whether the VGIIa-like strains displayed an increased mutation rate. We utilized a standard fluctuation assay approach to determine the rate of 5-FOA resistance, as previously applied in *Cryptococcus* (*Magditch et al., 2012*). The resistance rate was increased in the hypermutator lineage between 4.4-fold at the minimum (CBS7750) and 6.6-fold at the maximum (ICB107) compared to the type strain VGIIa R265 (*Figure 2A*). The majority of 5-FOA resistance in *Cryptococcus neoformans* is acquired through mutation of *URA5* (*Kwon-Chung et al., 1992*); the *URA5* locus was PCR amplified and sequenced in independently derived 5-FOA resistant isolates to determine the genetic basis of resistance. Mutations were characterized as either substitutions, or indels of either single or multiple bases (*Figure 2B*). The majority of resistant isolates were the result of substitutions, and the mutation profile was similar between the hypermutator VGIIa-like strains and the non-hypermutator VGIIa strain EJB17 (EJB17 vs NIH444: p=1.00, EJB17 vs CBS7750 p=0.60, EJB17 vs ICB107 p=1.00, Fisher's Exact Test).

## *MSH2* mutants accumulate indels in homopolymer runs

Next, the whole genome data was analyzed to ascertain whether the similarity in mutation spectrum observed in the *URA5* locus was recapitulated at multiple loci or if the *URA5* locus had unique properties. To do this, we focused on two relatively long branches within the *C. deuterogattii* phylogeny:

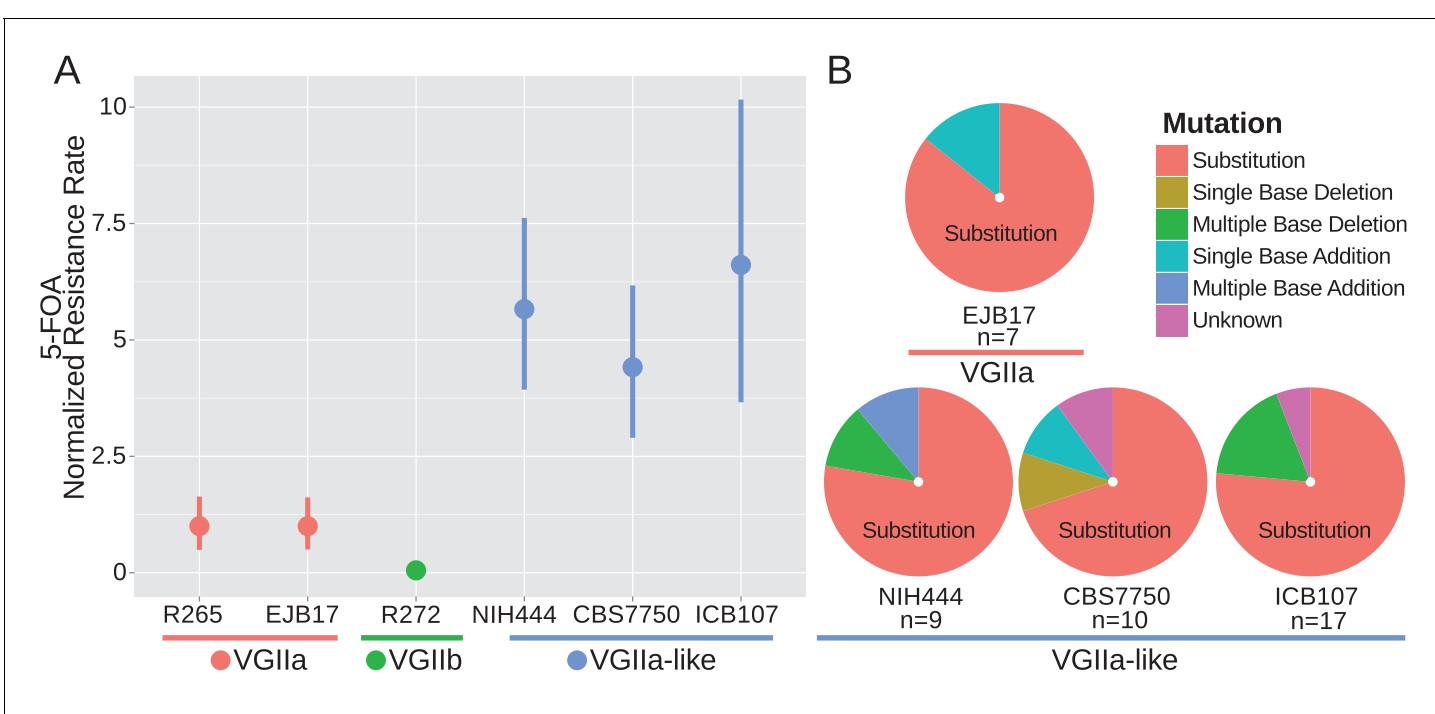

**Figure 2.** *msh2* mutants are hypermutators. (**A**) A fluctuation assay for resistance to 5-FOA was carried out for both wildtype VGIIa strains (R265 and EJB17) and hypermutator VGIIa-like strains (NIH444, EJB17, and ICB107), as well as an outgroup VGIIb (R272) strain. Resistance rates were normalized to the rate observed in R265. The VGIIa-like strains demonstrate an increase in mutation rate. Data shown are the mean of ten replicates with 95% confidence interval. (**B**) The molecular basis of resistance was determined for 5-FOA resistance at the *URA5* locus. All isolates tested demonstrated predominantly substitutions as the molecular basis of resistance.
DOI: https://doi.org/10.7554/eLife.28802.005

the branch separating the outgroup VGIIb strain R272 from the VGIIa clonal cluster (57,373 variants) and the private ICB107 variants, which represented the longest available hypermutator branch (719 variants). The frequency of individual SNPs was examined first. While a high rate of transitions as compared to transversions was maintained in the hypermutator branch, there was a slight reduction in the frequency of A->G and T->C mutations relative to C->T and G->A mutations (*Figure 3A*). However, much more striking was a dramatic increase in indels within homopolymer runs (*Figure 3B*). While shifts in homopolymer runs in the R272 branch account for only 0.84% of the variants, in ICB107 they account for 45.3%, exceeding even the proportion of SNPs.

By looking further at the context of the homopolymer run mutations within the ICB107 branch, it is apparent that longer base runs appear to be less stable than shorter runs, as approximately 24% of the mutations occur within the context of a nine base homopolymer run (*Figure 3C*). This peak is likely a function of both the decreased stability of longer runs and the low quantity of longer homopolymer runs in the genome. Many of these longer runs occur within intergenic regions in the genome, but a substantial portion of the coding genes in *C. deuterogattii* contain at least one longer coding homopolymer run (*Figure 3C*). The longest homopolymer run in the *URA5* locus contains only four bases, which is predicted to be both relatively stable and among the coding genes with relatively shorter homopolymer runs (*Figure 3C*). In contrast, the *FRR1* gene that encodes the FKBP12 homolog responsible for the antifungal action of both FK506 and rapamycin has a run of seven cytosines (7C) within its coding region. This places it both within the range of commonly mutated homopolymers in the ICB107 branch, and in the upper half of homopolymer containing genes in *C. deuterogattii*, suggesting that *FRR1* is an appropriate locus to test the effect of mismatch repair mutants on homopolymers containing genes (*Figure 3C*).

## Genes containing homopolymer runs are highly unstable in *msh2* mutants

The *FRR1* locus was utilized to further assess whether VGIIa-like strains were hypermutators at more than one locus and to test the hypothesis that homopolymer runs are particularly unstable in these isolates. The protein product of *FRR1*, FKBP12, binds to either FK506 or rapamycin to form a protein-drug complex that inhibits calcineurin or TOR, respectively. By selecting with both drugs at the same time at the non-permissive temperature of 37°C, loss of function mutations in *FRR1* are selected as the only single step mutation conferring resistance to both drugs. Here we utilized a semi-quantitative assay to identify gross differences in mutation rate whereby independent colonies were grown in liquid culture and then swabbed onto quadrants of a selective plate. On FK506/rapamycin media, the wildtype strains produced a small number of resistant colonies (*Figure 4A*). In contrast, the NIH444 hypermutator strain produced prolific FK506/rapamycin resistant colonies. Surprisingly, the CBS7750 hypermutator was completely resistant to both drugs. Upon examining the *FRR1* locus in the whole genome sequence data we discovered that there was already a single base deletion within the 7C homopolymer run in this strain. This may represent an unselected drug resistance phenotype potentiated by the *msh2* defect. We attempted to use this assay for the ICB107 strain as well but were unsuccessful, likely as a result of its inability to grow at the higher growth temperature necessary for this assay that is based upon calcineurin-dependent growth at 37°C (Figure 7B).

Next a standard fluctuation assay approach was employed to quantify the mutation rate difference between the VGIIa and VGIIa-like hypermutators. There was a greater than 120-fold increase in mutation rate to FK506/rapamycin resistance in NIH444 compared to R265, in contrast with the maximum of 6.6-fold increase observed for 5-FOA resistance (*Figure 4C*). As before, we selected independent resistant colonies, PCR amplified, and sequenced the *FRR1* gene to determine the mechanism of resistance. All resistance was explained by mutations within *FRR1* as expected, but while R265 still primarily acquired resistance through substitutions, NIH444 now almost exclusively underwent single base additions or single base deletions, all within the 7C homopolymer run (R265 vs NIH444, p=0.0003, Fisher's Exact Test). These indels resulted in nonsense mutations and resistance to both FK506 and rapamycin.

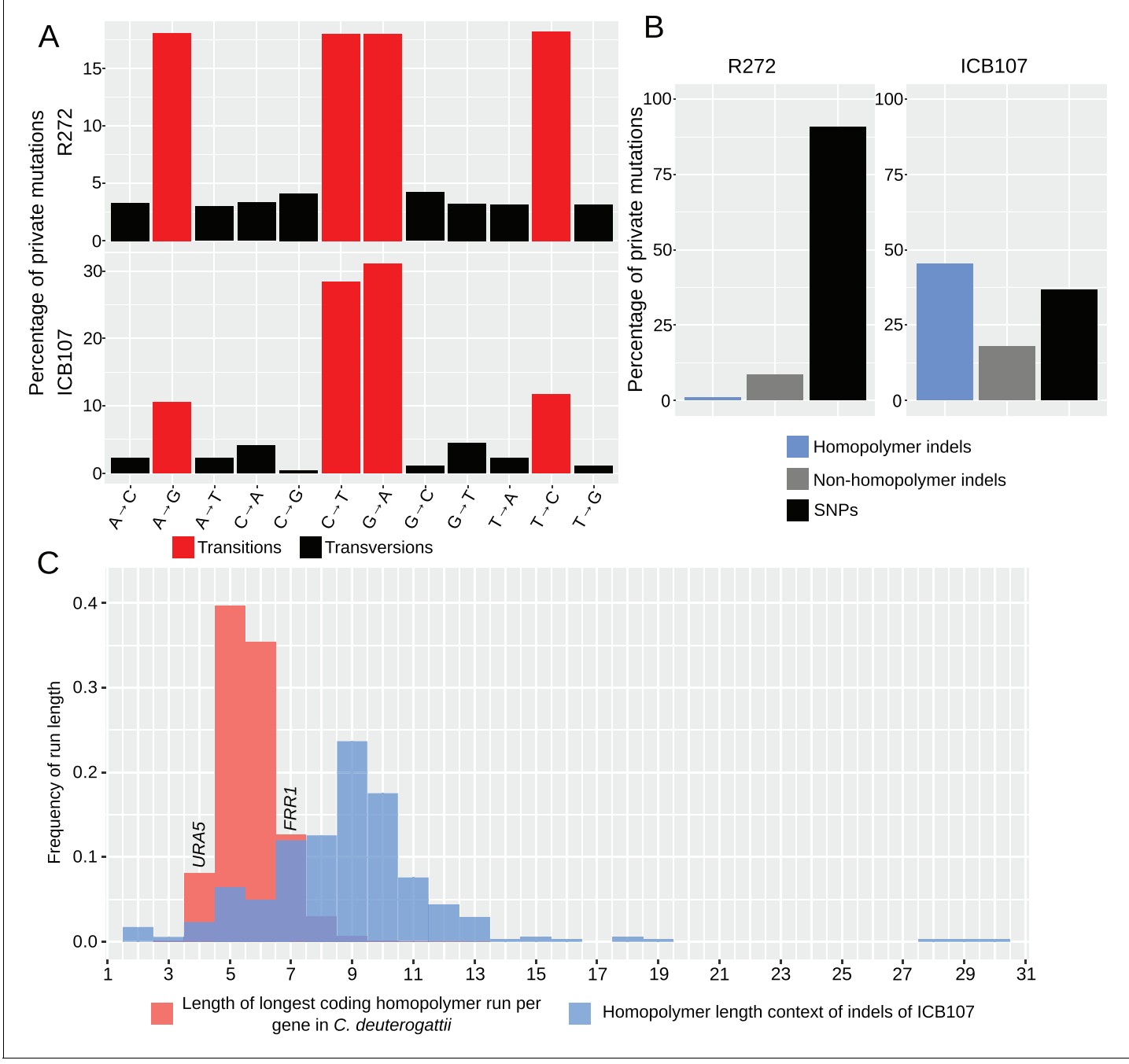

**Figure 3.** Hypermutators are characterized by high homopolymer instability. (**A**) Percentage of transitions and transversions among SNPs private to R272, an outgroup, and ICB107, a hypermutator, are shown. Transitions are more common in both strains, while one type of transition is modestly reduced in the hypermutator. (**B**) All private mutations within R272 and ICB107 were characterized either as SNPs, indels, or homopolymer indels. The cutoff used to distinguish a homopolymer indel was longer than four bases. ICB107 had a substantial increase in homopolymer run shifts. (**C**) Frequency of indels within all homopolymers longer than one base from the private ICB107 mutations are shown in blue. All genes from the *C. deuterogattii* transcriptome were grouped by frequency of the longest homopolymer run within the coding region of that gene and shown in red. *URA5* only has a run of 4 bases, while *FRR1*, which encodes FKBP12, the target of FK506 and rapamycin, has a run of 7 bases.

DOI: https://doi.org/10.7554/eLife.28802.006

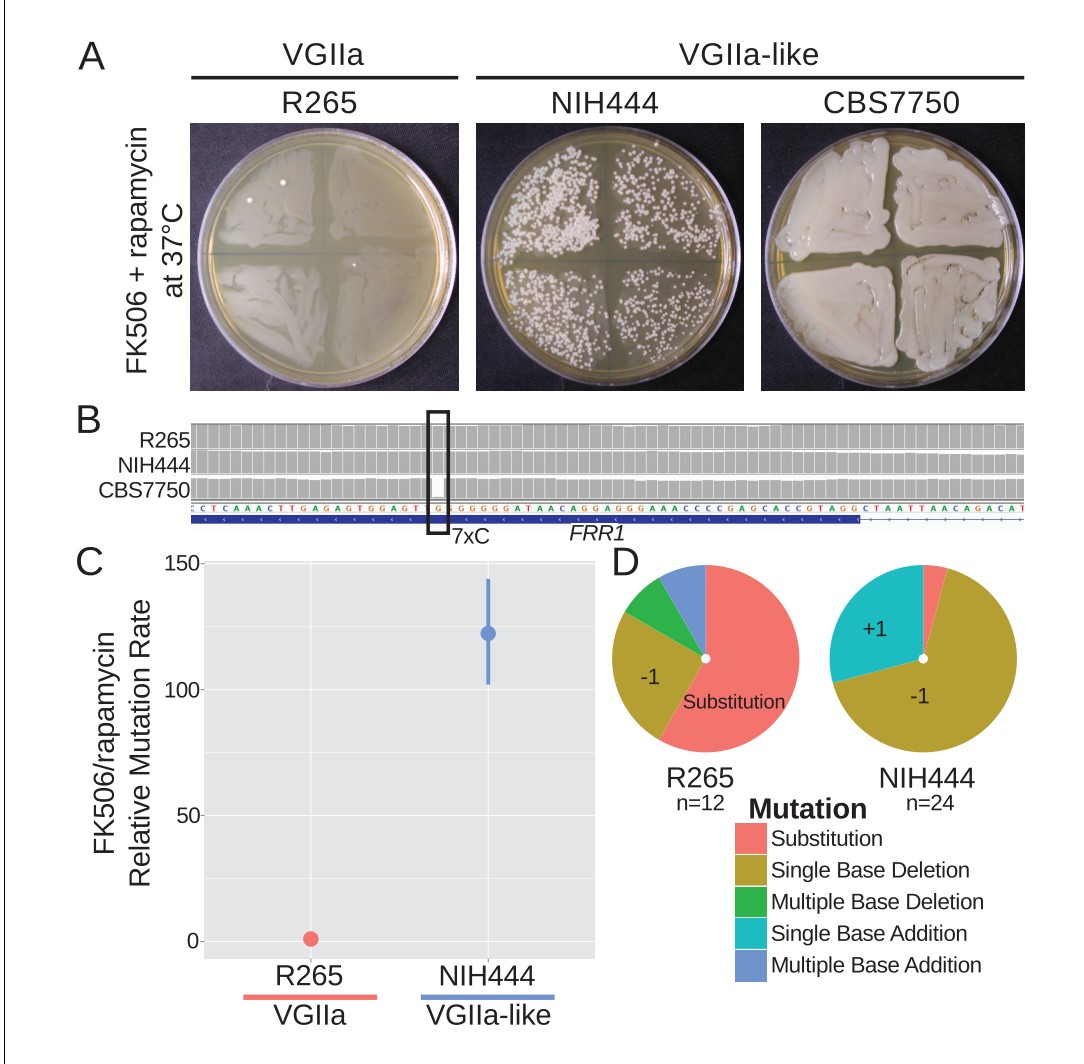

**Figure 4.** A homopolymer run within the *FRR1* gene allows rapid inactivation and resistance to FK506 and rapamycin. (**A**) Swab assays of the VGIIa and VGIIa-like strains were conducted to determine whether VGIIa-like stains develop resistance to FK506 and rapamycin at 37°C at an elevated rate compared to VGIIa. NIH444 demonstrated a large increase in resistance rate, while CBS7750 was completely resistant. (**B**) The resistance in CBS7750 is attributable to a single base deletion within the coding 7C run in the *FRR1* gene that has been fixed in this strain. (**C**) Fluctuation assays for NIH444 and R265 show that the hypermutator conferred a greater than 100-fold increase in mutation rate compared to the wildtype VGIIa strain. Data shown are the mean of ten replicates with 95% confidence intervals and are normalized to the rate in R265. (**D**) Analysis of the molecular basis of resistance shows that substitutions are still the predominant mechanism for resistance in R265, but in the hypermutator strain single base additions and single base deletions within the homopolymer run are responsible for the vast majority of resistance.
DOI: https://doi.org/10.7554/eLife.28802.007

## Hypermutation phenotype is linked to the *msh2* defect

Once we established a *bona fide* hypermutator phenotype in the VGIIa-like strains, we sought to verify that this phenotype was linked to the *msh2* del131 allele. We crossed the NIH444 strain to a very closely related R265**a** congenic strain recently generated by serial backcrossing (*Zhu et al., 2013*). *C. deuterogattii* matings are less fertile than *C. neoformans*, but we successfully dissected 14 viable spores from this cross. We typed these strains for the *MSH2* allele by sequencing and observed 1:1 segregation of the *MSH2* and the *msh2* del131 alleles. The spores were then tested for rate of FK506/rapamycin resistance using the same semi-quantitative swabbing assay described above (*Figure 5A*). While five of the *msh2* del131 strains displayed the predicted hypermutator phenotype,

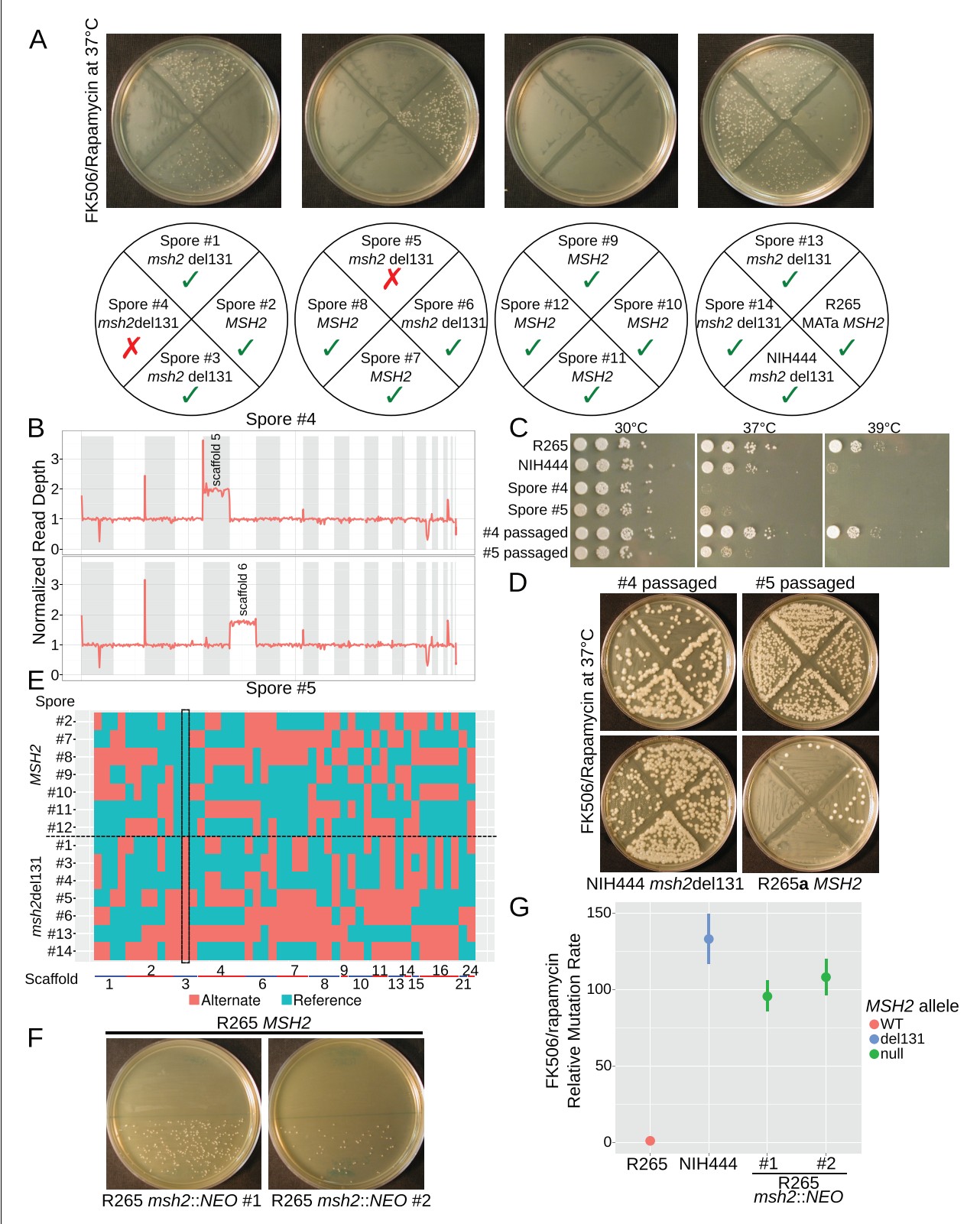

**Figure 5.** The hypermutator phenotype is linked to the frameshift in *MSH2*. (**A**) Progeny from a cross between NIH444 and R265 show co-segregation of the *msh2* del131 allele with the hypermutator phenotype in all except two cases. (**B**) Whole genome depth of coverage plots show that scaffolds five and six are aneuploid in spores #4 and #5, respectively. (**C**) Temperature sensitivity assay for the incongruent spores before and after serial passaging at 37°C. (**D**) After passage at 37°C, the non-congruent spores now properly demonstrate linkage with the *msh2* del131 allele. (**E**) Whole genome

*Figure 5 continued on next page*

*Figure 5 continued*

sequencing of NIH444 x R265a) progeny, with variant regions indicated as reference in blue or alternate in red. The boxed region indicates a SNP in close linkage to the *msh2* del131 allele. (F) Two independent de novo deletions of *MSH2* via biolistic transformation demonstrate elevated mutation rates in an FK506/rapamycin swabbing assay at 37°C. (G) Fluctuation assays for R265, NIH444, and two independent deletions of *msh2* in R265 show that the hypermutator phenotype is recapitulated in the null mutants. Data shown are the mean of ten replicates with 95% confidence intervals and are normalized to the rate observed in R265.

DOI: https://doi.org/10.7554/eLife.28802.008

The following source data and figure supplement are available for figure 5:

**Source data 1.** Aneuploidy in NIH444 x R265a cross.

DOI: https://doi.org/10.7554/eLife.28802.009

**Figure supplement 1.** Passaged spore progeny demonstrate linkage between *msh2*del131 allele and mutator phenotype.

DOI: https://doi.org/10.7554/eLife.28802.010

surprisingly two of the spores (#4 and 5) appeared wildtype despite inheritance of a defective copy of *MSH2*.

Mating in *Cryptococcus* has previously been demonstrated to produce phenotypic plasticity through the generation of aneuploid progeny at a high rate (*Ni et al., 2013*). These aneuploids often display defects in growth at high temperature; thus, we tested whether the high growth temperature of 37°C employed in our FK506/rapamycin resistance assay may have masked the ability of these meiotic progeny to develop drug resistance. Congruent with this hypothesis, both spore products #4 and #5 demonstrated growth defects at 37°C and 39°C (*Figure 5C*). Whole genome sequencing confirmed that both segregants carried an extra copy of one chromosome, suggesting that a 1N + 1 aneuploidy was causing temperature sensitivity (*Figure 5B*). In fact, all of the progeny from this mating had an unusually high rate of aneuploidy with 8 of the 14 progeny aneuploid for at least one scaffold/chromosome (*Figure 5—source data 1*). By passaging all of the spores on YPD at 37°C we were able to restore the ability to grow at high temperatures in all of the progeny after four to nine passages after which all no longer demonstrated a high temperature growth defect (*Figure 5C*). As a result the passaged derivatives of isolates #4 and #5 now demonstrated a hypermutator phenotype as expected (*Figure 5D*) and all 14 progeny produced the results predicted through linkage of the *msh2* defect to the hypermutator phenotype (*Figure 5—figure supplement 1*). Further, whole genome typing of the 14 meiotic progeny demonstrated that sufficient recombination had occurred to establish linkage. All 14 progeny demonstrated unique SNP profiles and only one SNP demonstrated inheritance congruent with the hypermutator phenotype: a SNP on scaffold 3 that was linked to the *msh2* del131 indel (~175 kb) (*Figure 5E*).

As a final verification of phenotypic linkage, we constructed two independent deletions of the *MSH2* gene in the R265 VGIIa background via biolistic transformation and homologous recombination replacing the *MSH2* ORF with a neomycin resistance cassette. A complete deletion of *MSH2* resulted in the same hypermutator phenotype as the *msh2* del131 allele, providing further evidence that the hypermutator phenotype was linked to the loss of function in *MSH2* (*Figure 5F*). A fluctuation assay was used to determine the mutation rate for resistance to FK506 and rapamycin and demonstrated that the null mutants did not have an elevated mutation rate in comparison to the NIH444 strain (*Figure 5G*). This suggests that suppressors have not arisen on the NIH444 background to moderate the effects of the *msh2* del131 allele.

## Hypermutation enables phenotypic diversity including reversion of mutations

During growth of the *msh2Δ::NEO* strains a spontaneous *ade2* mutant was fortuitously identified based on its classic red pigmentation (*Figure 6A*). We verified that this was the result of a defect in adenine biosynthesis as this strain grows on YNB supplemented with adenine, but not YNB media (*Figure 6B*). As expected, sequencing of the *ADE2* locus revealed that the ade⁻ phenotype was linked to a single base mutation that results in a His->Arg amino acid change (*Figure 6C*).

The red pigment produced by *ade2* mutants is a toxic intermediate in adenine synthesis that accumulates in the vacuoles of mutant cells. As a result, these mutants have a growth defect, and suppressor mutations can readily be isolated that eliminate production of this toxic intermediate and now produce white colonies (*Figure 6C*). We isolated two red and two white derivative colonies and

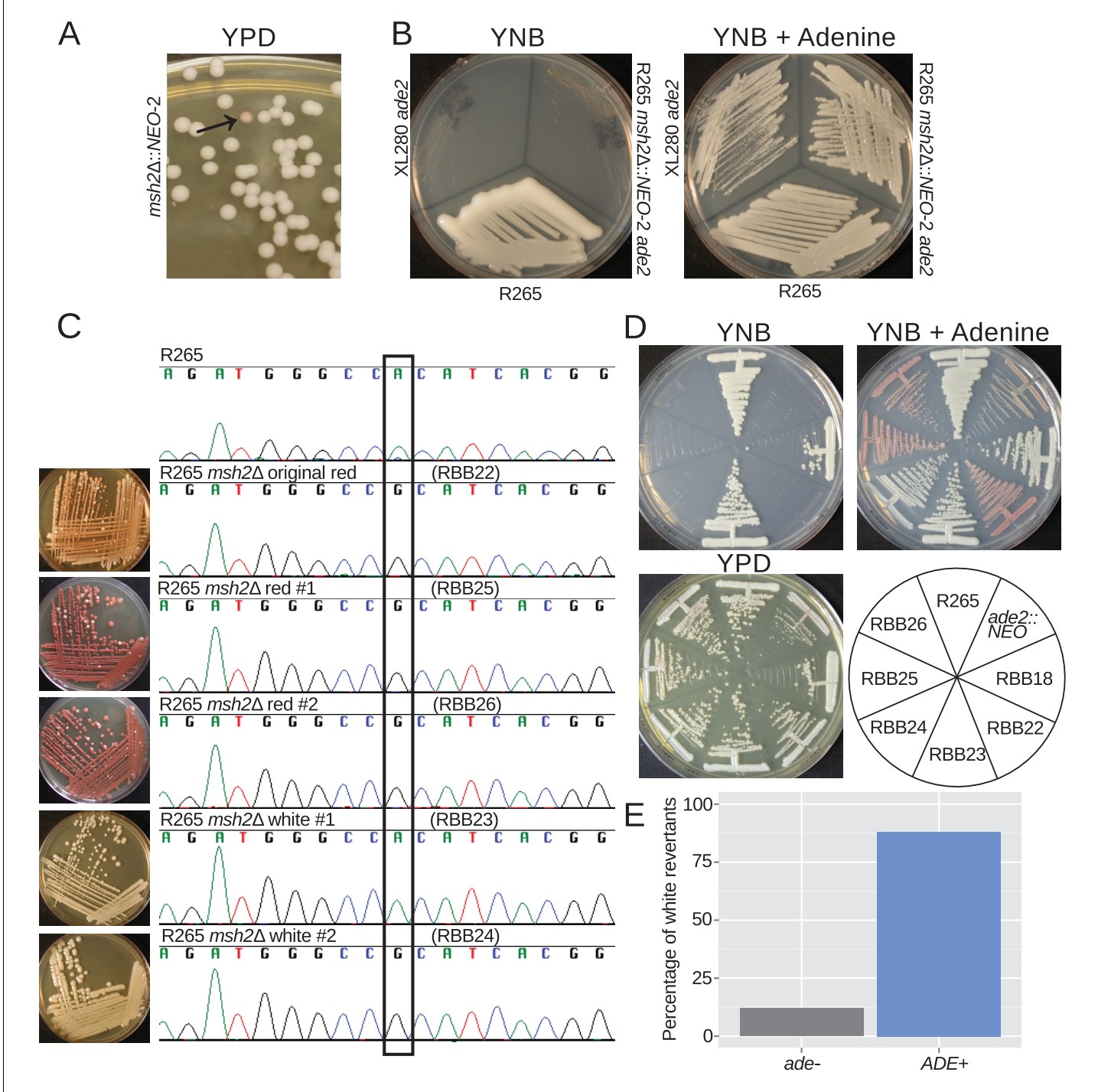

**Figure 6.** Hypermutation allows inactivation and reactivation of adenine biosynthetic pathway. (**A**) A spontaneous red colony was isolated from the de novo *msh2* deletion (RBB18). (**B**) This colony (RBB22) demonstrated adenine auxotrophy, suggesting that it was an *ade2* mutant. (**C**) Sequencing of the *ADE2* locus confirmed that the original colony was an *ade2* mutant. In addition, two red (RBB25 and RBB26) and two white derivatives (RBB23 and RBB24) were tested. One white derivative had reverted the original mutation (RBB23), while the second had eliminated production of the red intermediate but had not reverted the original *ade2* mutation (RBB24). (**D**) One revertant colony (RBB23) demonstrated adenine prototrophy, while the other (RBB24) remained an auxotroph despite losing the red pigmentation. (**E**) An assay to test direct reversion frequency versus secondary mutation to eliminate the red toxic intermediate demonstrated that the most common mutations were direct reversions and restoration of adenine prototrophy.

DOI: https://doi.org/10.7554/eLife.28802.011

The following figure supplement is available for figure 6:

*Figure 6 continued on next page*

*Figure 6 continued*

**Figure supplement 1.** Reversion of *ade2* mutants primarily occurs through repair of *ADE2*.
DOI: https://doi.org/10.7554/eLife.28802.012

sequenced the *ADE2* locus. While the red colonies retained the causative substitution, one of the white colonies retained it and the other had undergone reversion at this site back to the functional nucleotide (*Figure 6C*). We confirmed that the revertant isolate was no longer an adenine auxotroph and that the second white isolate was still auxotrophic, despite the lack of pigmentation (*Figure 6D*). This is likely the result of a second mutation upstream in the adenine biosynthetic pathway, resulting in further inactivation and loss of the ability to produce the toxic red intermediate. We tested a number of independent white reverted colonies for the ability to grow on media lacking adenine and observed that direct reversion to functional adenine biosynthesis was more common than additional inactivating mutations, with 37/42 (88%) white colonies demonstrating adenine prototrophy (*Figure 6E*). Taken together, these results suggest that hypermutation allows both frequent inactivation of pathways, and also the means to either adapt to the consequences or to simply directly revert the original mutation. In the context of antifungal drug action or activity, this phenotypic switching could be highly important.

## Hypermutation does not directly affect virulence

We previously demonstrated that the VGIIa-like mutants exhibit diminished virulence in a nasal instillation model of virulence (*Fraser et al., 2005*; *Byrnes et al., 2010*). Here, we tested whether loss of Msh2 function was directly responsible for the decrease in virulence but observed no difference in virulence between R265 and two independent de novo deletions of *msh2* (*Figure 7A*). This suggested that the defects in murine virulence in the VGIIa-like isolates could be the result of multiple independent deleterious mutations unique to each lineage. We next tested the ability to grow at higher temperature, a critical virulence factor, in the VGIIa-like isolates. All three isolates showed defects in high temperature growth, varying from minor defects in NIH444, the isolate with the shortest branch to R265, to moderate defects in CBS7750, the isolate with an intermediate length branch, and finally severe defects in ICB107, the isolate with the longest branch affected by the hypermutator (*Figure 7B*). This suggests that instead of an immediate change in virulence, defects in mismatch repair may instead result in loss of virulence over time as mutations accumulate in critical pathways. Thus, rather than the virulence difference being explained by a single change or set of changes shared by all three VGIIa-like strains, instead they may all be avirulent for their own unique reasons.

## Hypermutation is deleterious in rich conditions, but advantageous under stress

Our data suggests that over time a high mutation rate comes at a cost to organisms. We next addressed the early stages of hypermutator emergence and the direct effects of Msh2 inactivation to address if temporary benefits may be conferred. To this end, independent *msh2 de novo* deletions were employed in competition experiments with wildtype R265. Briefly, liquid YPD cultures were mixed in 50:50 ratios of wildtype R265 and a neomycin resistant *msh2* mutant or a random insertion of the neomycin resistance cassette as a control. After 48 hr incubation, co-cultures were spread onto YPD plates and then colonies were picked and restruck to media containing neomycin to determine the percentage of colonies derived from each original strain. If mutations are neutral, the expectation is that the 50:50 ratio will be maintained. Instead, growth defects were observed for the *msh2* mutants at either 30°C or 37°C, suggesting that the *msh2* mutation is deleterious under rich growth conditions (*Figure 7C*). Notably, in three of the 24 replicates at 30°C or 37°C the *msh2* mutant was able to grow moderately better than the wildtype parent strain. This may indicate that in these individual experimental replicates the hypermutator allowed a beneficial mutation and hitchhiked to higher frequency. However, under highly stressful conditions, such as exposure to the drugs FK506/rapamycin at 37°C, the hypermutator was highly advantageous. In multiple replicates the hypermutator rapidly acquired resistance and overtook the entire co-culture. This suggests that mutator alleles can be advantageous when a population faces an evolutionary landscape where large

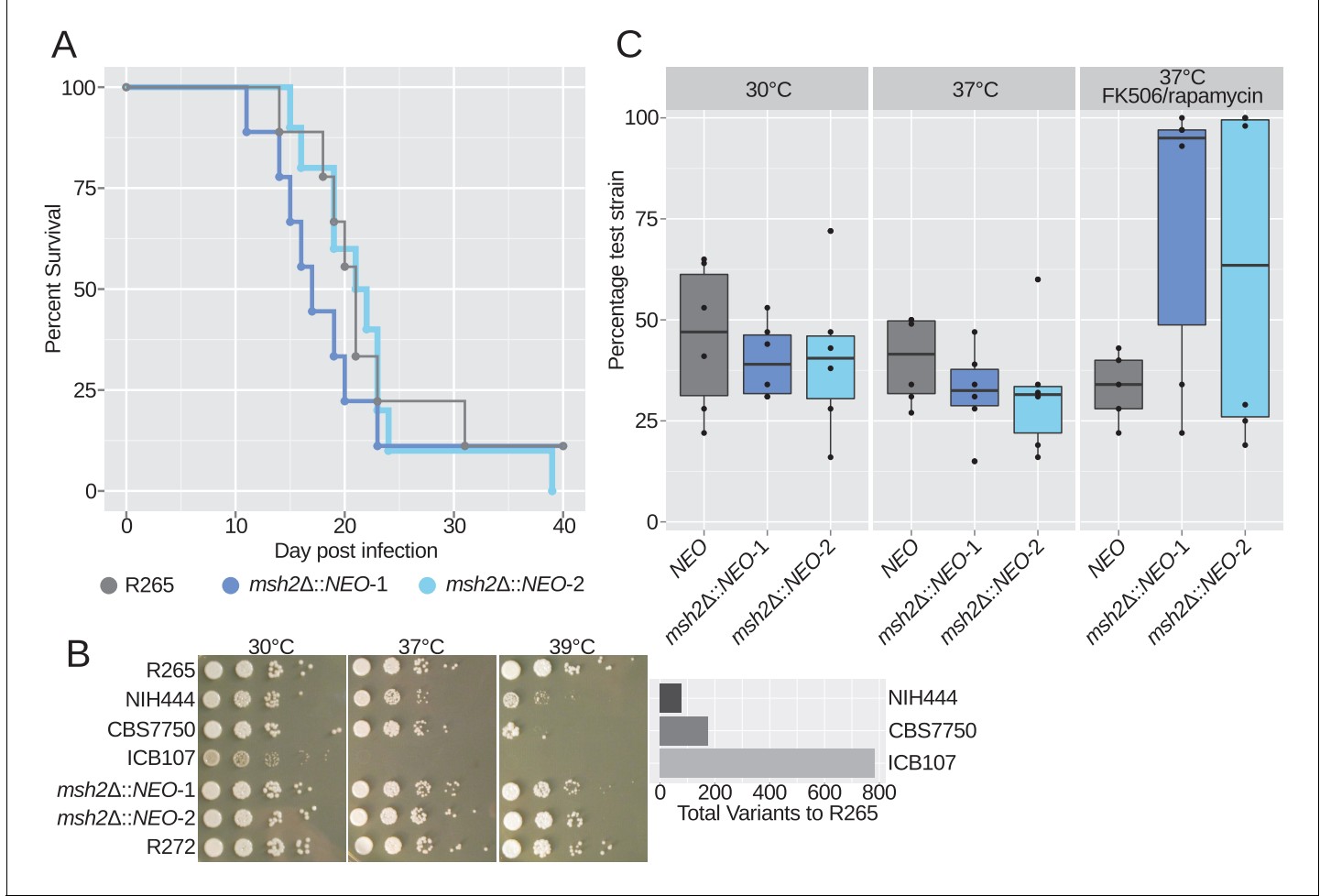

**Figure 7.** Hypermutation does not have immediate virulence defects but may potentiate long term deficits. (**A**) Virulence tested in the murine inhalation model was not strongly affected by deletion of the *MSH2* gene. (**B**) All three VGIIa-like strains demonstrated defects in high temperature growth via a spot dilution assay in comparison with the VGIIa R265 strain and the VGIIb R272 strain as an outgroup. Strains with longer branches exhibited larger high temperature growth defects. (**C**) Competition experiments between a tester strain with the neomycin resistance marker and the wildtype R265 strain. (Strain used: SEC501, RBB17, RBB18). Original cultures were mixed in a 1:1 ratio and then grown overnight in liquid YPD. Both hypermutators showed a modest growth defect at 30°C and 37°C but a dramatic growth advantage in the high stress FK506/rapamycin 37°C condition. Boxplots show minimum, first quartile, median, third quartile, and maximum values. Points represent the results from six individual replicates summarized by the box plot. The *NEO* vs WT competition is shown in gray, while the two *msh2Δ::NEO* competitions are shown in dark and light blue.
DOI: https://doi.org/10.7554/eLife.28802.013

effect beneficial mutations exist. In contrast, mutators are deleterious in landscapes where few large effect mutations can provide advantages. Antifungal treatment and transitions from environment to host are likely to provide opportunities for adaptive mutation and favor mutator alleles.

## Hypermutator lineage is derived and hypermutation did not cause the VGIIa outbreak

Historically, *C. deuterogattii* has been thought of as a tropical and subtropical pathogen. As a result, the origin of the VGIIa outbreak in the Pacific Northwest, a non-tropical environment, was surprising. In addition, (1) the South American origin of the ICB107 strain (within the proposed cradle of the *C. deuterogattii* species [*Hagen et al., 2013*]), (2) the isolation of NIH444 in Seattle near the outbreak origin, and (3) the diminished virulence of the VGIIa-like subclade (*Fraser et al., 2005*; *Byrnes et al., 2010*), all combined to suggest that the VGIIa-like group might have been the immediate precursors to the clonal VGIIa outbreak cluster (*Billmyre et al., 2014*; *Engelthaler et al., 2014*). Identification of the hypermutator phenotype further suggested that the defect in Msh2, carried by the older

VGIIa-like group, may have played a role in adaptation to the climate of the Pacific Northwest and may have even potentiated the increase in virulence. To test this, a maximum parsimony phylogeny of the VGIIa-like and VGIIa strains was constructed, including additional sequences for a related South American VGII clade not included in our previous analysis (*Engelthaler et al., 2014*) and using the VGIIb R272 strain as an outgroup (*Figure 8*). As described above, the *msh2* mutation can allow restoration of function mutations at the exact mutation site that are indistinguishable from the original sequence (*Figure 6*). Likewise, mating could also reintroduce a functional *MSH2* allele. For these reasons, we tested for the possible presence and impact of the past *msh2* del131 mutator allele throughout the phylogeny by examining the imputed mutation spectrum.

As discussed above (*Figure 3*), branches with defects in mismatch repair show an increased frequency of shifts in homopolymer runs. Increases in homopolymer run shifts were observed only on the most proximal branch ancestral to the VGIIa-like mutator lineage. This suggests that the mutator phenotype is congruent with the presence of the allele throughout the VGIIa-like phylogeny, and that the VGIIa group did not experience a transient period of *msh2* mediated hypermutation, followed by repair or mating-mediated replacement. Instead, the VGIIa-like lineage may represent a unique pathway to adaptation distinct from that followed by the VGIIa group that resulted in diminished rather than enhanced virulence.

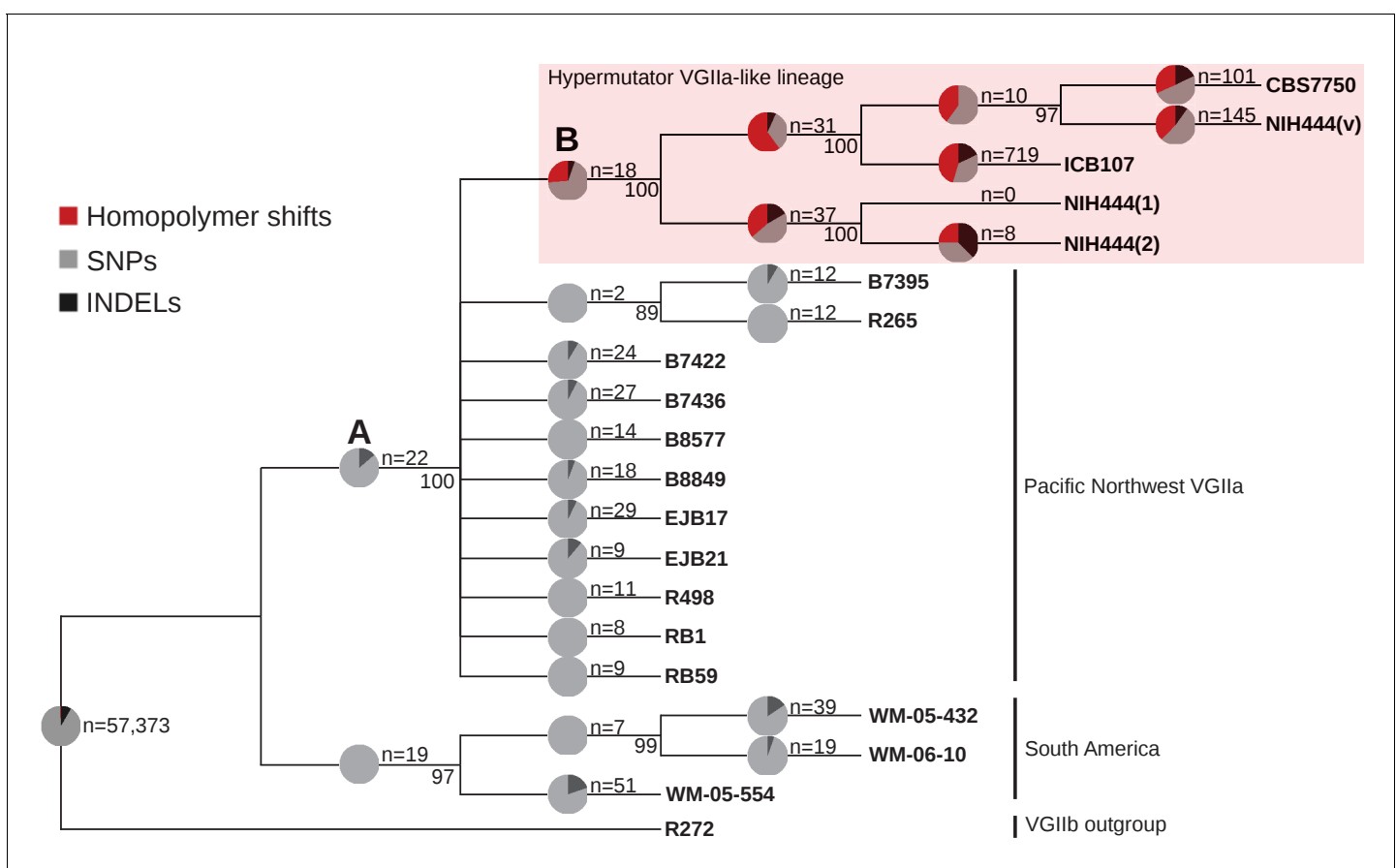

**Figure 8.** The VGIIa-like hypermutator is derived and not ancestral to the Pacific Northwest outbreak. A maximum parsimony phylogeny of the VGIIa group with the VGIIb R272 genome as an outgroup demonstrates that the VGIIa-like group is a branch parallel to the VGIIa group. To test for the presence of a defect in MMR throughout the tree, the mutation spectrum was examined on each branch. High rates of homopolymer run shifts were observed throughout the VGIIa-like group, but no evidence was apparent at branch A). Instead it appears that the hypermutator first arose on branch B).

DOI: https://doi.org/10.7554/eLife.28802.014

## Discussion

In this study we have identified and characterized a successful lineage of eukaryotic hypermutators. Elevated mutations rates are a common adaptive mechanism in bacteria, but are typically thought of as transient states that allow beneficial mutations in the short term but are selected against in the long term (*Wielgoss et al., 2013*). Bacteria solve this problem through horizontal gene transfer of genes from the mismatch repair pathway, allowing the initial beneficial mutation to be separated from the deleterious mutator allele (*Denamur et al., 2000*). In contrast, few cases have been identified in natural isolates of eukaryotes, suggesting that variation in mutation rate may play a less substantial role than in bacteria. Recent studies have identified exceptions to this rule. In *S. cerevisiae*, incompatible alleles of *PMS1* and *MLH1* result in elevated mutation rates if present in combination (*Heck et al., 2006*; *Bui et al., 2015*). However, in the rare cases where the incompatible arrangement is found in nature, additional suppressors have arisen to restore wildtype mutation rate, suggesting that hypermutator phenotypes may not be tolerated over long periods of time (*Bui et al., 2017*; *Skelly et al., 2017*). In addition, a recent study of *Candida glabrata* has demonstrated that a substantial proportion of clinical isolates (>50%) carry nonsynonymous mutations in the *MSH2* gene, some causing elevated mutation rates similar to the null phenotype and others with intermediate changes in mutation rate (*Healey et al., 2016*). The authors correlate the mismatch repair defects with multi-drug resistance. A second study with a different sample cohort confirmed the presence of *MSH2* mutations, but concluded that drug resistance was better correlated with drug exposure than with mismatch repair defects (*Dellière et al., 2016*).

In contrast with the studies in *S. cerevisiae* and *C. glabrata*, here we identified a group of viable hypermutator strains resulting from a nonsense mutation in *MSH2*. These strains were isolated over a period of fifteen years from two different continents and include both clinical and environmental strains. Increases in mutation rate were similar, although slightly lower in magnitude, to those observed previously in the model yeast *S. cerevisiae*, where *msh2* mutations increase resistance rate to Canavanine by 40 fold compared to 6.8 fold for 5-FOA here, or reversion of a 7xT run in a *HOM3* mutant by 662 fold compared to 120 fold for 7xC in *FRR1* here (*Marsischky et al., 1996*). The mutation spectrum was also similar, as homopolymer run indels were responsible almost exclusively for mutations within run-containing genes in *msh2* mutants, but not in wildtype strains as previously reported in *S. cerevisiae* (*Tran et al., 1997*). The *msh2* del131 allele carried by these strains appears to result in a complete loss of function, rather than simply reducing the efficiency of mismatch repair. This is distinct from the mutators identified in *C. glabrata*, and suggests that the VGIIa-like lineage is a successful and relatively long-lived hypermutator lineage, capable of both disseminating over a large area and persisting in the environment. *Cryptococcus deuterogattii* may also represent an intermediate between mutator-rich *C. glabrata* and mutator-poor *S. cerevisiae*. *C. deuterogatttii* is not an obligate pathogen and can cycle between an environmental lifestyle and an infectious lifestyle as an 'accidental pathogen' (*Springer et al., 2012*). Elevated mutation rates may be selected for by Red Queen interactions with a host, meaning that pathogens like *C. glabrata* that grow only in association with their hosts would experience higher mutation rates than facultative pathogens like *C. deuterogattii*, or effectively nonpathogenic yeasts like *S. cerevisiae*. In fact, a putative recurrent infection of *Cryptococcus neoformans* was recently demonstrated to contain nonsense mutations in *MSH2*, *MSH5*, and *RAD5*, predicted to result in a mutator phenotype (*Rhodes et al., 2017*).

In addition, a second recent study also identified two other hypermutator strains from clinical isolates of *C. neoformans* as well (*Boyce et al., 2017*). Boyce *et al.* went on to examine changes in mutation rate and spectrum with similar results to those presented here, although with a much larger induction of mutation rate in *URA5* than reported here (around 200 fold vs 6.8 fold). This could reflect differences between DNA repair in *C. neoformans* and *C. deuterogattii*. They also observed changes in stability of homopolymer runs, similar to those described here but in a second homopolymer-containing locus. Interestingly, this study also observed loss of ability to grow at high temperatures in one of three replicates of *msh2* mutants in an experimental evolution assay, like that observed here in the VGIIa-like hypermutator clade. Notable as well is that hypermutators in *C. neoformans* have thus far been observed only in clinical isolates, even in studies that included matched environmental sample populations like Boyce et al. This provides further evidence that microbial lifestyle may be affecting the frequency of mutator strains, with pathogens being more likely to evolve elevated mutation rate, perhaps in response to drug or host challenges.

An alternate hypothesis is that ploidy may play a role in the success of mutator strains. *C. deuterogattii, C. neoformans,* and *C. glabrata* exist primarily as haploids in nature, while *S. cerevisiae* is predominantly a diploid. Diploid strains are buffered from the effects of loss of function mutations like those observed in homopolymer runs in this study, which may reduce the supply of large effect beneficial mutations for mutator alleles to hitchhike upon in diploids. However, past work in *S. cerevisiae* suggests that diploids adapt more quickly in *msh2* mutant backgrounds, rather than less, suggesting that ploidy may play the opposite role, at least in *S. cerevisiae* (*Thompson et al., 2006*).

Evolutionary theory predicts that the admixture provided by sex in a population nullifies the ability of mutator alleles to hitchhike to high frequency (*Tenaillon et al., 2000*). Unlike obligately sexual animals, or asexual bacteria, fungi can reproduce both sexually and asexually, with the frequency of sex varying substantially between different species. We previously described *C. deuterogattii* as a species characterized by long periods of mitotic clonal expansion and only intermittent sexual crosses (*Billmyre et al., 2014*). A contibuting factor is likely the highly biased mating type distribution in *C. deuterogattii,* as the vast majority of *C. deuterogattii* isolates are *MATα*, with only a handful of *MAT**a*** isolates described globally (*Byrnes et al., 2010*). In *Cryptococcus deneoformans,* which shares a similar biased mating type distribution, this has resulted in the development of a unisexual α-α sexual cycle, dispensing with the obligate need for a *MAT**a*** partner (*Lin et al., 2005*). This unisexual cycle can result in both de novo variation, but also recombination and admixture at similar levels to that observed in typical bisexual crosses (*Ni et al., 2013*; *Lin et al., 2005*; *Sun et al., 2014*). However, no laboratory unisexual cycle has been observed in *C. deuterogattii* to this point. Consequently, as strains are introduced to new locales, they may need to survive and adapt via mitotic growth without sexual crosses for long periods of time. This could elevate linkage between mutator alleles and the beneficial mutations they elicit, but also eliminate the ability to separate those beneficial mutations from additional deleterious alleles.

We also showed that hypermutators can allow both inactivating mutations in genes, and reversion of those mutations to the wildtype. Mismatch repair-defective mutator strains are characterized by particularly high rates of slippage within homopolymer runs, and these occur at run lengths that are common within fungal genomes. This may indicate that eukaryotes may harbor contingency loci featuring homopolymer runs, like those observed in bacteria (*Moxon et al., 2006*). Fungal contingency loci could be critical for responses to antifungal agents, both in the environment and the clinical setting. In addition, variation could be important in host-pathogen interactions as well. For example, in *S. cerevisiae*, tandem repeats are enriched within cell surface genes, which could enable alteration of antigenic diversity during pathogenic interactions (*Verstrepen et al., 2005*). While these loci are unstable even in wildtype populations, defects in the MMR pathway could enhance this instability and result in increased diversity. In *C. neoformans,* a mutator phenotype has been previously described in a lab-passaged strain notable for frequent mutations in the RAM pathway, which enables a dimorphic transition between pseudohyphal and yeast phase growth, although the molecular basis of the elevated mutation rate is unknown (*Magditch et al., 2012*). This transition is highly important for survival in the face of environmental threats such as amoeba, but compromises pathways responsible for high temperature growth, suggesting that oscillation between RAM+ and RAM- states may allow populations to survive as conditions change.

In addition, we provided evidence that the VGIIa-like lineage was not a subvirulent progenitor of the Pacific Northwest outbreak, in contrast with previous hypotheses from two groups (*Billmyre et al., 2014*; *Engelthaler et al., 2014*). Rather, the VGIIa-like lineage is derived and may represent either an alternative pathway of adaptation or potentially an accelerated stage of the evolutionary trajectory of the outbreak. The ability to grow at high temperatures is a critical component of virulence for fungal pathogens. However, many pathogenic fungi are thought to be 'accidental' pathogens where virulence factors are selected for by other environmental factors. In this model, thermotolerance could be selected by high ambient temperatures in a non-host environment (*Casadevall et al., 2003*). Moving from a tropical/sub-tropical environment in South America to the more temperate climate of the Pacific Northwest may relieve this high temperature selection. Consequently, the loss of ability to grow under thermal stress observed in the VGIIa-like strains may be adaptive for the majority of their growth conditions in the environment, but also render them relatively avirulent in mammals. This could suggest that as the VGIIa outbreak strains continue to adapt to the Pacific Northwest, they will gradually lose virulence potential, like the VGIIa-like lineage, making the outbreak self-limiting over time. Alternatively, the defects could simply reflect a decrease in

viability caused by a long period of growth with an elevated mutation rate, resulting in mutational meltdown. The most severe temperature defect was observed in ICB107, a strain isolated from a patient in Brazil and also the strain with the largest number of mutations separating it from the VGIIa group, which would support the second hypothesis. These strains may have been able to persist in the Pacific Northwest in spite of the high mutation load because of a bottleneck effect during their introduction.

Our original hypothesis was that the common variants that differentiated the VGIIa group from the progenitor of the VGIIa-like mutator group would explain the decline in virulence observed in the VGIIa-like strains (*Billmyre et al., 2014*; *Byrnes et al., 2010*). However, the *msh2* mutation, the only predicted large effect mutation, has no obvious immediate effect on virulence in a murine inhalation model. This is interesting, at least in part, because *msh2* mutants of *C. neoformans* were previously reported to exhibit a modest growth advantage in mouse lungs in pooled signature tagged mutant experiments (*Liu et al., 2008*). This suggests that the mutator strains are better able to adapt to growth in the mouse lung, but not in a way that correlates directly with virulence in a dissemination and survival model of virulence. Boyce et al. obtained similar results in *C. neoformans* to those reported here, with *msh2Δ* and *mlh1Δ* mutants appearing neutral in a murine infection model, and *pms1Δ* mutants even relatively less virulent (*Boyce et al., 2017*). In addition, our results suggest that loss of virulence in the VGIIa-like strains may be a consequence of independent private mutations, i.e. each VGIIa-like strain is avirulent for its own reason. In previous work, CBS7750 and ICB107 have the most substantial virulence defects, while NIH444 has a more modest defect (*Byrnes et al., 2010*). This is congruent with the fact that NIH444 has the shortest divergence from the VGIIa clonal group, while CBS7750 and ICB107 have much longer branches, and is also congruent with the high temperature growth defects we observed.

Finally, we demonstrated that mutation spectrum analysis could be utilized within a phylogeny to infer the presence of a hypermutator allele through an increase in the proportion of indels within homopolymer runs. In bacteria, the hypermutator state is often transient and wildtype mismatch repair is often restored via horizontal gene transfer (*Denamur et al., 2000*). This paradigm could also operate in fungi, with sexual recombination replacing direct DNA transfer. However, these events are difficult to detect if the hypermutator allele is purged from the population by selection after it is separated from its linked beneficial allele. The widespread availability of whole genome sequencing could now allow changes in mutation spectrum to be used to detect episodic hypermutation throughout entire populations of microbes. We suspect that these episodes are common throughout the evolution of eukaryotic microbes and may be even more common among pathogenic microbes, reflecting their natural history as well as the result of Red Queen host-pathogen conflicts.

## Materials and methods

### Strains and media

Strains used in this study are listed in *Supplementary file 1*. Primers used are listed in *Supplementary file 2*. Strains were routinely grown on YPD media at 30°C and maintained in permanent glycerol stocks at −80°C. Strains marked with neomycin resistance were grown on YPD supplemented with G418.

### Genetic crosses and spore dissection

To conduct crosses and isolate spores, NIH444 was cocultured with the R265**a** congenic strain (*Zhu et al., 2013*) on solid V8 pH = 5.0 medium in the dark for eight weeks. Basidiospores were isolated using a microdissection microscope equipped with a 25 μm needle (Cora Styles Needles 'N Blocks, Dissection Needle Kit) as previously described (*Hsueh et al., 2006*).

### Virulence assays

Approximately eight week old A/J mice were anesthetized with phenobarbital via intraperitoneal injection and then were infected with $5 \times 10^4$ cells from strains R265, RBB17, and RBB18 by intranasal inhalation. Ten animals per group were infected. Mice were monitored daily for signs of cryptococcal infection and sacrificed when exhibiting signs of clinical distress. The animal study was conducted in the Division of Laboratory Animal Resources (DLAR) facilities at Duke University

Medical Center (DUMC). All of the animal work was performed according to the guidelines of NIH and Duke University Institutional Animal Care and Use Committee (IACUC) under protocol number A245-13-09.

## Gene disruption and strain construction

Deletions of the *MSH2* gene were constructed using a standard overlap PCR approach as previously described (*Davidson et al., 2000*). Briefly, 1 kb flanking regions of genomic DNA were amplified from both the 5' and 3' regions flanking the *MSH2* open reading frame in R265 and the selectable marker for *NEO* resistance was amplified from plasmid pJAF1. An overlap PCR was carried out to generate a full-length deletion cassette and R265 was transformed using biolistic transformation. Gene replacement was confirmed using in-gene, 5' junction, 3' junction, and spanning PCRs. Two independent transformations of independent overnight cultures of R265 were carried out to isolate independent mutants.

The R265 *NEO* resistance marked strain was constructed by transforming XbaI digested pJAF1 plasmid into wild-type R265 by biolistics. Transformants were selected for on YPD containing neomycin. Transformants were checked for tandem insertions using primers JOHE40500/JOHE40501 and a strain with only a single insertion was chosen and employed for the competition assay.

## Fluctuation assays

Fluctuation assays were performed on either synthetic medium containing 5-FOA (1 g/L) at 30°C or YPD supplemented with rapamycin (1 μg/mL) and FK506 (1 μg/mL) at 37°C. For each strain tested, ten independent 5 mL YPD liquid cultures were grown overnight at 30°C. Cultures were then split and either spread directly on selective media or diluted and spread on solid YPD to determine colony forming units. Resistant colonies and total colonies were counted and mutation rate was calculated using the Maximum Likelihood method as implemented via the FALCOR calculator (*Hall et al., 2009*).

## Co-culture competition assays

Competition assays were carried out by growing independent liquid cultures overnight in 5 mL YPD. Cultures were then counted using a hemocytometer and 500,000 cells each of a tester neomycin resistant strain and a wildtype R265 culture were mixed in a 5 mL liquid YPD culture. Co-cultures were grown for 48 hr and then spread onto a YPD plate, such that individual colonies could be isolated. All colonies were picked (up to 60 and at least 5) and restruck to neomycin media to determine the proportion of colony forming units derived from each of the original strains in the competition.

## Genome sequencing and assembly

DNA was isolated with the CTAB isolation protocol as previously described (*Pitkin et al., 1996*). Library construction and genome sequencing were carried out at the University of North Carolina Next Generation Sequencing Facility. Paired-end libraries with approximately 300-base inserts were constructed and 100 base reads were generated A number of genome sequences were previously available (*Billmyre et al., 2014*; *Engelthaler et al., 2014*). The remaining were generated here and are deposited on the SRA under project accession no. PRJNA387047. All sequences were mapped to the V2 R265 reference genome (*Farrer et al., 2015*). Alignment was performed using BWA MEM with default settings (*Li and Durbin, 2009*). Further processing was carried out using the Genome Analysis Toolkit (GATK) version 3.8 (*McKenna et al., 2010*), including SAMtools (*Li et al., 2009*) and Picard. SNPs and indels were called with the HaplotypeCaller from GATK, using the haploid setting. GATK's VariantFiltration module was used to hard filter variants as suggested using the following expressions: 'QD <2.0 || FS >60.0 || MQ <40.0 " for SNPs and 'QD <2.0 || FS >200.0' for indels. GATK's VariantAnnotator was used to define the homopolymer context of indels. VCFtools was used to filter sites with missing data and extract private variants (*Danecek et al., 2011*). Variants resulting from erroneous calls in repetitive regions were manually removed. SnpEff was employed to determine the predicted impact of mutations (*Cingolani et al., 2012*).

### Inference of phylogeny

Maximum parsimony phylogenies were constructed using MEGA6 (*Tamura et al., 2013*) to analyze SNP matrices extracted from VCF format using a custom Perl script (*Source code file 1*). SNPs and indels were placed on branches where they were predicted to change states using VCFtools (*Danecek et al., 2011*) to extract the variants supporting each node.

## Acknowledgements

Supported by NIH/NIAID R37 grant AI39115-20 and R01 grant AI50113-13 to JH. We thank Sheng Sun, Shelby Priest, Paul Magwene, and Thomas Petes for helpful discussion and suggestions during the writing of this manuscript.

## Additional information

### Funding

| Funder | Grant reference number | Author |
| --- | --- | --- |
| National Institute of Allergy and Infectious Diseases | R37 AI39115-20 | Joseph Heitman |
| National Institute of Allergy and Infectious Diseases | R01 AI50113-13 | Joseph Heitman |

The funders had no role in study design, data collection and interpretation, or the decision to submit the work for publication.

### Author contributions

R Blake Billmyre, Conceptualization, Data curation, Software, Formal analysis, Validation, Investigation, Visualization, Methodology, Writing—original draft, Writing—review and editing; Shelly Applen Clancey, Formal analysis, Validation, Investigation, Methodology, Writing—review and editing; Joseph Heitman, Conceptualization, Resources, Formal analysis, Supervision, Funding acquisition, Methodology, Writing—original draft, Project administration, Writing—review and editing

### Author ORCIDs

R Blake Billmyre https://orcid.org/0000-0003-4866-3711

### Ethics

Animal experimentation: The animal study was conducted in the Division of Laboratory Animal Resources (DLAR) facilities at Duke University Medical Center (DUMC). All of the animal work was performed according to the guidelines of NIH and Duke University Institutional Animal Care and Use Committee (IACUC) under protocol number A245-13-09. Mice were monitored daily for signs of cryptococcal infection and sacrificed when exhibiting signs of clinical distress.

### Decision letter and Author response

Decision letter https://doi.org/10.7554/eLife.28802.026
Author response https://doi.org/10.7554/eLife.28802.027

## Additional files

### Supplementary files

• Source code file 1. Custom Perl script
DOI: https://doi.org/10.7554/eLife.28802.015

• Supplementary file 1. Strains used in this study.
DOI: https://doi.org/10.7554/eLife.28802.016

• Supplementary file 2. Oligonucleotides used in this study.

DOI: https://doi.org/10.7554/eLife.28802.017

• Supplementary file 3. SNP data used to build phylogeny

DOI: https://doi.org/10.7554/eLife.28802.018

• Transparent reporting form

DOI: https://doi.org/10.7554/eLife.28802.019

## Major datasets

The following dataset was generated:

| Author(s) | Year | Dataset title | Dataset URL | Database, license, and accessibility information |
|---|---|---|---|---|
| Billmyre RB, Clancey SA, Heitman J | 2017 | Cryptococcus gattii VGII Genome sequencing | https://www.ncbi.nlm.nih.gov/bioproject/PRJNA387047/ | Publicly available at the NCBI BioProject database (accession no: PRJNA387047) |

The following previously published datasets were used:

| Author(s) | Year | Dataset title | Dataset URL | Database, license, and accessibility information |
|---|---|---|---|---|
| Billmyre RB, Croll D, Li W, Mieczkowski P, Carter DA, Cuomo CA, Kronstad JW, Heitman J | 2014 | Cryptococcus gattii VGIII | https://www.ncbi.nlm.nih.gov/bioproject/PRJNA254979 | Publicly available at the NCBI BioProject database (accession no. PRJNA254979) |
| Engelthaler DM, Hicks ND, Gillece JD, Roe CC, Schupp JM, Driebe EM, Gilgado F, Carriconde F, Trilles L, Firactive C, Ngamskulrungroj P, Castaneda E, Lazera MDS, Melhem MSC, Perez-Bercoff A, Huttley G, Sorrell TC, Voelz K, May RC, Fisher MC, Thompson GR, Lockhart SR, Keim P, Meyer W | 2014 | Cryptococcus gattii VGIII | https://www.ncbi.nlm.nih.gov/bioproject/?term=PRJNA244927 | Publicly available at the NCBI BioProject database (accession no. PRJNA244927) |

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
