## [Decision Letter]

Thank you for submitting your article "Natural mismatch repair mutations mediate phenotypic diversity and drug resistance in *Cryptococcus deuterogattii*" for consideration by *eLife*. Your article has been reviewed by three peer reviewers, one of whom is a member of our Board of Reviewing Editors, and the evaluation has been overseen by Diethard Tautz as the Senior Editor. The reviewers have opted to remain anonymous.

The reviewers have discussed the reviews with one another and the Reviewing Editor has drafted this decision to help you prepare a revised submission.

Summary:

In this nicely executed study, Billmyre and coworkers identified three closely related isolates of the VGIIa outbreak of the haploid fungal pathogen *Cryptococcus deuterogatti*, each of which contained the same frameshift mutation in the msh2 mismatch repair gene. The work focuses on the characterization of the msh2 mutants, which they show to yield a hypermutator phenotype (i.e., increased mutation rate). This work characterizes in detail the hypermutator phenotype and its effects on sequence diversity (e.g., hypermutation results in accumulation of indels in homopolymer runs) as well as on phenotypic diversity (e.g., they showed in competition assays that the msh2 mutation conferred a selective advantage under stress, but not in rich growth conditions). In addition, they used a phylogeny of the outbreak and related isolates to argue that the hypermutator phenotype did not cause the VGIIa outbreak. Finally, they concluded that the hypermutation phenotype was likely to "result in a loss of virulence over time as mutations accumulate in critical pathways."

The novel part of this work is the finding that *Cryptococcus deuterogattii* isolates have been found in nature that bear a defective DNA mismatch repair gene and display a mutator phenotype. Population genetic theory predicts that diploid organisms that undergo sex would not maintain mutators, because of fitness costs and removal of linkage between a mutator allele and a beneficial mutation. Natural isolates of prokaryotes that are mutators have been frequently observed, but at present there appears to be only two other examples in eukaryotes (in tumor cells – Healy et al., 2016 – and, in *Cryptococcus* – Rhodes et al., 2017 and the recent study by Boyce et al. in mBio). The authors hypothesize that mutators were identified in *C. deuterogatti* because this organism primarily grows as a haploid and appears to undergo long periods of clonal expansion.

Essential revisions:

1) The authors should incorporate in the discussion of the manuscript the recent work by Boyce and colleagues (https://doi.org/10.1128/mBio.00595-17) on *Cryptococcus neoformans*. The study is very similar in premise to the work described in this manuscript, but because some of the experiments performed in the two studies are different as well as because the two organisms studied are closely related, the two manuscripts nicely complement each other. For example, the study by Boyce et al. examines the genetic impact of the hypermutator phenotype through very well done allele-swap experiments, and details the impact of these genotypes on clinically-important phenotypes such as drug resistance and virulence. It would be great to present and discuss these experiments in the context of this work, as they strengthen the case for the existence of hypermutators in fungal, and more generally eukaryotic, pathogens.

2) The existence of environmental MSH2 hypermutators in *Cryptococcus* species was also previously (recently) shown by Rhodes et al. The authors cite this study in their Discussion – they should also mention it in the Introduction of their manuscript.

---

## [Author Response]

Essential revisions:1) The authors should incorporate in the discussion of the manuscript the recent work by Boyce and colleagues (https://doi.org/10.1128/mBio.00595-17) on Cryptococcus neoformans. The study is very similar in premise to the work described in this manuscript, but because some of the experiments performed in the two studies are different as well as because the two organisms studied are closely related, the two manuscripts nicely complement each other. For example, the study by Boyce et al. examines the genetic impact of the hypermutator phenotype through very well done allele-swap experiments, and details the impact of these genotypes on clinically-important phenotypes such as drug resistance and virulence. It would be great to present and discuss these experiments in the context of this work, as they strengthen the case for the existence of hypermutators in fungal, and more generally eukaryotic, pathogens.

We thank the reviewers for this suggestion. We did not discuss this manuscript in our initial submission because it was not yet published and we were unaware of the content or that it had been submitted. We agree that it is nicely complementary to our finding of hypermutators in *C. deuterogattii* by extending the finding of hypermutators to the sibling species of *C. neoformans*, and as such will add to our understanding of the presence of hypermutators in fungal pathogens. We have explicitly mentioned and cited the Boyce et al. manuscript now in the revised Introduction and also introduced more substantial commentary in the revised Discussion.

2) The existence of environmental MSH2 hypermutators in Cryptococcus species was also previously (recently) shown by Rhodes et al. The authors cite this study in their Discussion – they should also mention it in the Introduction of their manuscript.

We have added text referring to this manuscript in the revised Introduction as well.